# DenoiSeg: Joint Denoising and Segmentation

**Abstract.** Microscopy image analysis often requires the segmentation of objects, but training data for this task is typically scarce and hard to obtain. Here we propose DenoiSeg, a new method that can be trained end-to-end on only a few annotated ground truth segmentations. We achieve this by extending Noise2Void, a self-supervised denoising scheme that can be trained on noisy images alone, to also predict dense 3-class segmentations. The reason for the success of our method is that segmentation can profit from denoising, especially when performed jointly within the same network. The network becomes a denoising expert by seeing all available raw data, while co-learning to segment, even if only a few segmentation labels are available. This hypothesis is additionally fueled by our observation that the best segmentation results on high quality (very low noise) raw data are obtained when moderate amounts of synthetic noise are added. This renders the denoising-task non-trivial and unleashes the desired co-learning effect. We believe that DenoiSeg offers a viable way to circumvent the tremendous hunger for high quality training data and effectively enables learning of dense segmentations when only very limited amounts of segmentation labels are available.

**Keywords:** segmentation · denoising · co-learning

## 1 Introduction

The advent of modern microscopy techniques has enabled the routine investigation of biological processes at sub-cellular resolution. The growing amount of microscopy image data necessitates the development of automated analysis methods, with object segmentation often being one of the desired analyses. Over the years, a sheer endless array of methods have been proposed for segmentation [9], but deep learning (DL) based approaches are currently best performing [3, 14, 18]. Still, even the best existing methods offer plenty of scope for improvements, motivating further research in this field [21, 23, 7].

A trait common to virtually all DL-based segmentation methods is their requirement for tremendous amounts of labeled ground truth (GT) training

---
\* Equal contribution (alphabetical order).

**Fig. 1.** The proposed DENOISEG training scheme. A U-NET is trained with a joint self-supervised denoising loss ($\mathcal{L}_d$) and a classical segmentation loss ($\mathcal{L}_s$). Both losses are weighted with respect to each other by a hyperparameter $\alpha$. In this example, $\mathcal{L}_d$ can be computed on all $m = 3800$ training patches, while $\mathcal{L}_s$ can only be computed on the $n = 10$ annotated ground truth patches that are available for segmentation.

data, the creation of which is extraordinarily time consuming. In order to make the most out of a given amount of segmentation training data, data augmentation [22, 27] is used in most cases. Another way to increase the amount of available training data for segmentation is to synthetically generate it, *e.g.* by using Generative Adversarial Networks (GANs) [8, 15, 20]. However, the generated training data needs to capture all statistical properties of the real data and the respective generated labels, thereby making this approach cumbersome in its own right.

For other image processing tasks, such as denoising [12, 26, 2], the annotation problem has been addressed via self-supervised training [10, 1, 11, 17]. While previous denoising approaches [26] require pairs of noisy and clean ground truth training images, self-supervised methods can be trained directly on the noisy raw data that is to be denoised.

Very recently, Prakash *et al.* [16] demonstrated on various microscopy datasets that self-supervised denoising [10] prior to object segmentation leads to greatly improved segmentation results, especially when only small numbers of segmentation GT images are available for training. The advantage of this approach stems from the fact that the self-supervised denoising module can be trained on the full body of available microscopy data. In this way, the subsequent segmentation module receives images that are easier to interpret, leading to an overall gain in segmentation quality even without having a lot of GT data to train on. In the context of natural images, a similar combination of denoising and segmentation was proposed by Liu *et al.* [13] and Wang *et al.* [25]. However, both methods lean heavily on the availability of paired low- and high-quality image pairs for training their respective denoising module. Additionally, their cascaded denoising and segmentation networks make the training comparatively computationally expensive.

Here, we present DENOISEG, a novel training scheme that leverages denoising for object segmentation (see Fig. 1). Like Prakash *et al.*, we employ self-supervised NOISE2VOID [10] for denoising. However, while Prakash *et al.* rely on two sequential steps for denoising and segmentation, we propose to use a single network to jointly predict the denoised image and the desired object segmentation. We use a simple U-NET [19] architecture, making training fast and

accessible on moderately priced consumer hardware. Our network is trained on noisy microscopy data and requires only a small fraction of images to be annotated with GT segmentations. We evaluate our method on different datasets and with different amounts of annotated training images. When only small amounts of annotated training data are available, our method consistently outperforms not only networks trained purely for segmentation [4, 6], but also the currently best performing training schemes proposed by Prakash *et al.* [16].

## 2    Methods

We propose to jointly train a single U-NET for segmentation and denoising tasks. While for segmentation only a small amount of annotated GT labels are available, the self-supervised denoising module benefits from all available raw images. In the following we will first discuss how these tasks can be addressed separately and then introduce a joint loss function combining the two.

**Segmentation.** We see segmentation as a 3-class pixel classification problem [4, 6, 16] and train a U-NET to classify each pixel as foreground, background or border (this yields superior results compared to a simple classification into foreground and background [21]). Our network uses three output channels to predict each pixel's probability of belonging to the respective class. We train it using the standard cross-entropy loss, which will be denoted as $\mathcal{L}_s\big(\boldsymbol{y}_i, f(\boldsymbol{x}_i)\big)$, where $\boldsymbol{x}_i$ is the $i$-th training image, $\boldsymbol{y}_i$ is the ground truth 3-class segmentation, and $f(\boldsymbol{x}_i)$ is the network output.

**Self-Supervised Denoising.** We use the NOISE2VOID setup described in [10] as our self-supervised denoiser of choice. We extend the above mentioned 3-class segmentation U-NET by adding a fourth output channel, which is used for denoising and trained using the NOISE2VOID scheme. NOISE2VOID uses a Mean Squared Error (MSE) loss, which is calculated over a randomly selected subset of blind spot pixels that are masked in the input image. Since the method is self-supervised and does not require ground truth, this loss $\mathcal{L}_d\big(\boldsymbol{x}_i, f(\boldsymbol{x}_i)\big)$ can be calculated as a function of the input image $\boldsymbol{x}_i$ and the network output $f(\boldsymbol{x}_i)$.

**Joint-Loss.** To jointly train our network for denoising and segmentation we use a combined loss. For a given training batch $(\boldsymbol{x}_1, \boldsymbol{y}_1, \ldots, \boldsymbol{x}_m, \boldsymbol{y}_m)$ of $m$ images, we assume that GT segmentation is available only for a subset of $n \ll m$ raw images. We define $\boldsymbol{y}_i = \boldsymbol{0}$ for images where no segmentation GT is present. The loss over a batch is calculated as

$$\mathcal{L} = \frac{1}{m} \sum_{i=1}^{m} \alpha \cdot \mathcal{L}_d\big(\boldsymbol{x}_i, f(\boldsymbol{x}_i)\big) + (1 - \alpha) \cdot \mathcal{L}_s\big(\boldsymbol{y}_i, f(\boldsymbol{x}_i)\big), \qquad (1)$$

where $0 \leq \alpha \leq 1$ is a tunable hyperparameter that determines the relative weight of denoising and segmentation during training. Note that the NOISE2VOID loss is

self-supervised, therefore it can be calculated for all raw images in the batch. The cross-entropy loss however requires GT segmentation and can only be evaluated on a subset of images, where this information is available. For images where no GT segmentation is available we define $\mathcal{L}_s\big(\boldsymbol{y}_i = \boldsymbol{0}, f(\boldsymbol{x}_i)\big) = 0$.

In the setup described above, setting $\alpha = 1$ corresponds to pure NOISE2VOID denoising. However, setting $\alpha = 0$ does not exactly correspond to the vanilla 3-class segmentation, due to two reasons. Firstly, only some of the images are annotated but in Eq. 1 the loss is divided by the constant batch size $m$. This effectively corresponds to a reduced batch size and learning rate, compared to the vanilla method. Secondly, our method applies NOISE2VOID masking of blind spot pixels in the input image.

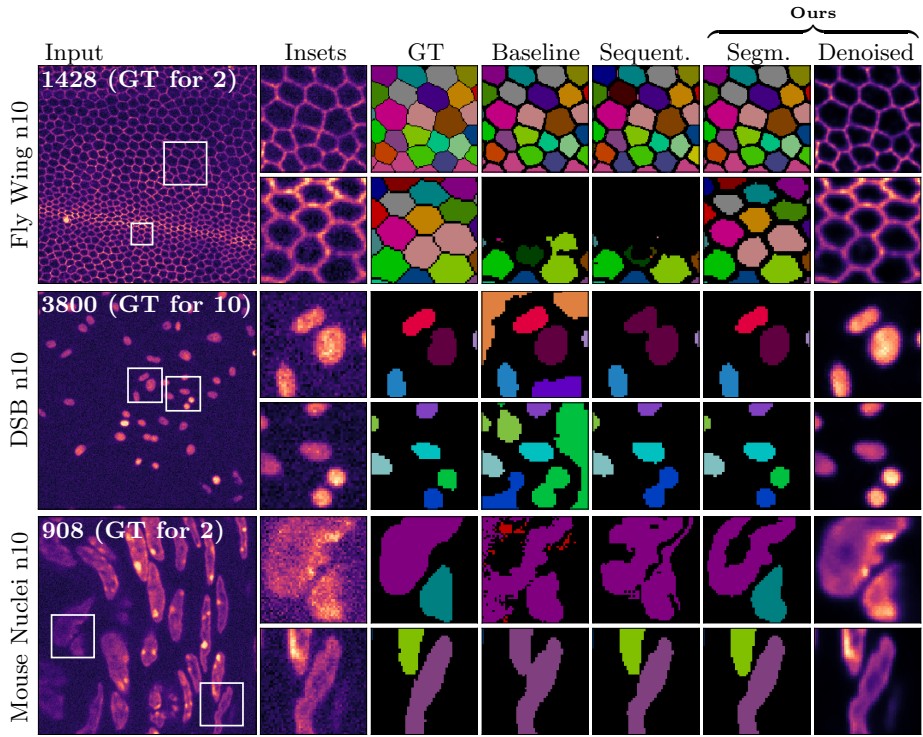

**Fig. 2.** Qualitative results on Fly Wing n10 (first row), DSB n10 (second row) and Mouse Nuclei n10 (third row). The first column shows an example test image. Numbers indicate how many noisy input and annotated ground truth (GT) patches were used for training. Note that segmentation GT was only available for at most 10 images, accounting for less than 0.27% of the available raw data. Other columns show depicted inset regions, from left to right showing: raw input, segmentation GT, results of two baseline methods, and our DENOISEG segmentation and denoising results.

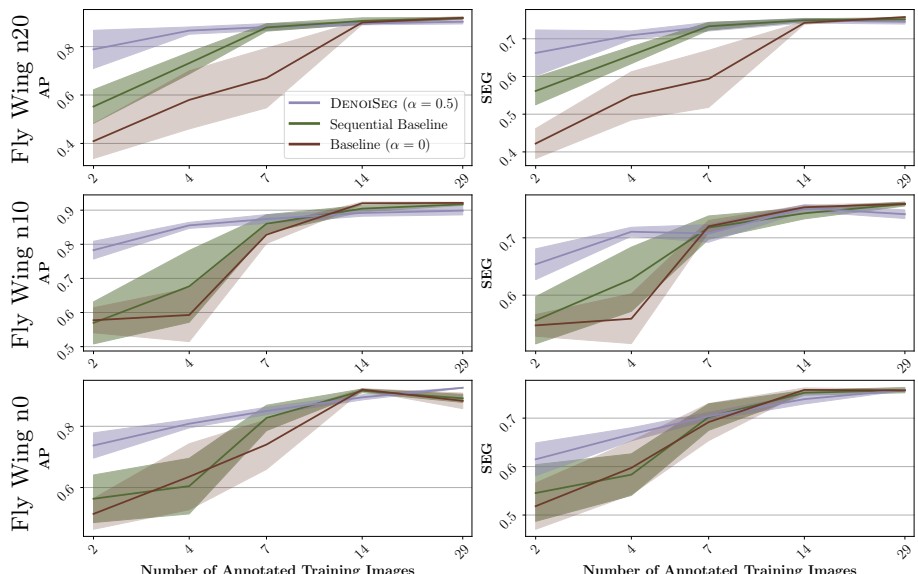

**Fig. 3.** Results for Fly Wing n0, n10 and n20, evaluated with Average Precision (AP) [21] and SEG-Score [24]. DENOISEG outperforms both baseline methods, mainly when only limited segmentation ground truth is available.

**Implementation Details.** Our DENOISEG implementation is publicly available[1]. The proposed network produces four output channels corresponding to denoised images, foreground, background and border segmentation. For all our experiments we use a U-NET architecture of depth 4, convolution kernel size of 3, a linear activation function in the last layer, 32 initial feature maps, and batch normalization during training. All networks are trained for 200 epochs with an initial learning rate of 0.0004. The learning rate is reduced if the validation loss is not decreasing over ten epochs. For training we use 8-fold data augmentation by adding 90° rotated and flipped versions of all images.

## 3   Experiments and Results

We use three publicly available datasets for which GT annotations are available (data available at DENOISEG-Wiki[2]). For each dataset we generate noisy versions by adding pixel-wise independent Gaussian noise with zero-mean and standard deviations of 10 and 20. The dataset names are extended by n0, n10, and n20 to indicate the respective additional noise. For network training, patches of size $128 \times 128$ are extracted and randomly split into training (85%) and validation (15%) sets.

---

[1] https://github.com/******/*******
[2] https://github.com/******/*******/wiki

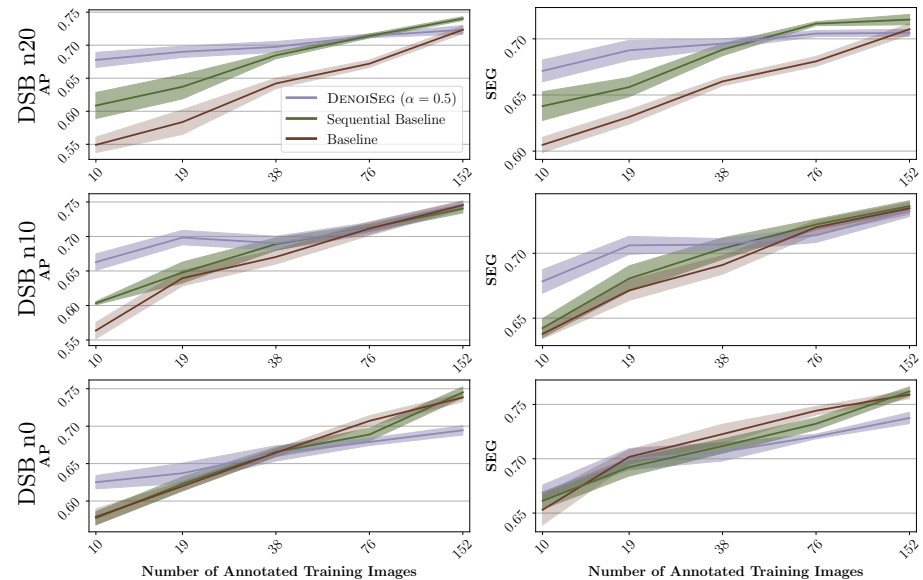

**Fig. 4.** Results for DSB n0, n10 and n20, evaluated with Average Precision (AP) [21] and SEG-Score [24]. DENOISEG outperforms both baseline methods, mainly when only limited segmentation ground truth is available. Note that the advantage of our proposed method is at least partially compromised when the image data is not noisy (row 3).

– **Fly Wing.** This dataset from our collaborators consist of 1428 training and 252 validation patches of a membrane labeled fly wing. The test set is comprised of 50 additional images.
– **DSB.** From the Kaggle 2018 Data Science Bowl challenge, we take the same images as used by [16]. The training and validation sets consist of 3800 and 670 patches respectively, while the test set counts 50 images.
– **Mouse Nuclei.** Finally, we choose a challenging dataset depicting diverse and non-uniformly clustered nuclei in the mouse skull, consisting of 908 training and 160 validation patches. The test set counts 67 additional images.

For each dataset, we train DENOISEG and compare it to two different competing methods: DENOISEG trained purely for segmentation with $\alpha = 0$ (referred to as *Baseline*), and a sequential scheme based on [16] that first trains a denoiser and then the aforementioned baseline (referred to as *Sequential*). We chose our network with $\alpha = 0$ as baseline to mitigate the effect of batch normalization on the learning rate as described in Section 2. A comparison of our baseline to a vanilla 3-class U-NET with the same hyperparameters leads to very similar results and can be found in Appendix B. Furthermore, we investigate DENOISEG performance when trained with different amounts of available GT segmentation images. This is done by picking random subsets of various sizes from the available GT annotations. Note that the self-supervised denoising task still has access to

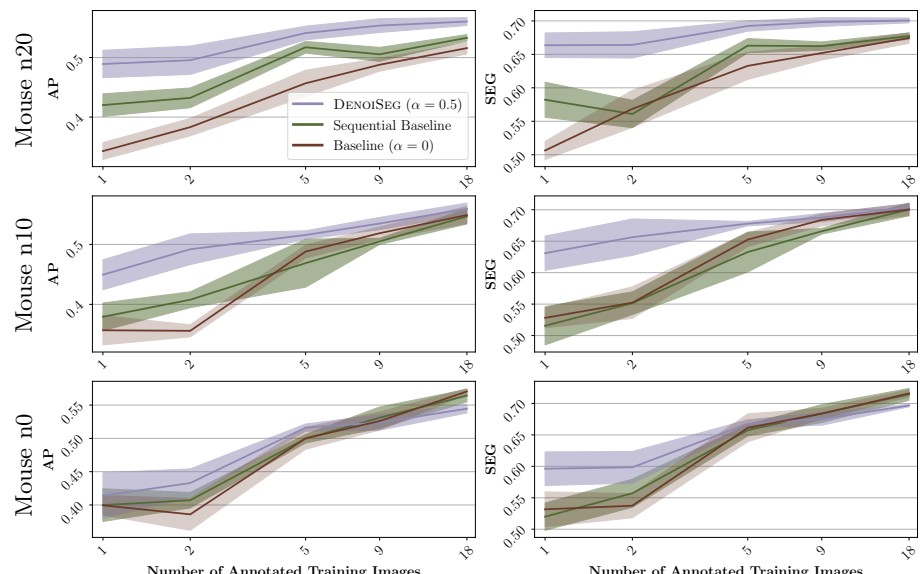

**Fig. 5.** Results for Mouse nuclei n0, n10 and n20, evaluated with Average Precision (AP) [21] and SEG-Score [24]. DenoiSeg outperforms both baseline methods, mainly when only limited segmentation ground truth is available.

all raw input images. A qualitative comparison of DenoiSeg results with other baselines (see Figure 2) indicates the effectiveness of our method.

As evaluation metrics, we use Average Precision (AP) [5] and SEG [24] scores. The AP metric measures both instance detection and segmentation accuracy while SEG captures the degree of overlap between instance segmentations and GT. To compute the scores, the predicted foreground channel is thresholded and connected components are interpreted as instance segmentations. The threshold values are optimized for each measure on the validation data. All conducted experiments were repeated 5 times and the mean scores along with $\pm 1$ standard error of the mean are reported in Figure 8.

**Performance with Varying Quantities of GT Data and Noise.** Figure 3 shows the results of DenoiSeg with $\alpha = 0.5$ (equally weighting denoising and segmentation losses) for Fly Wing n0, n10 and n20 datasets. For low numbers of GT training images, DenoiSeg outperforms all other methods. Similar results are seen for the other two datasets (see Figure 4 and Figure 5). Results for all performed experiments showing overall similar trends and can be found on the DenoiSeg-Wiki.

**Importance of $\alpha$.** We further investigated the sensitivity of our results to the hyperparameter $\alpha$. In Figure 6(a) we look at the segmentation performance of different values for hyperparameter $\alpha$. We compare the results of $\alpha = 0.3$ and $\alpha =$

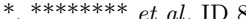

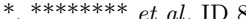

**Fig. 6.** In **(a)**, we show that DENOISEG consistently improves results over the baseline for a broad range of hyperparameter $\alpha$ values by looking at the difference $\Delta$ of AP to $\alpha = 0.5$. The results come close to what would be achievable by choosing the best possible $\alpha$ (see main text). In **(b)**, we show that adding synthetic noise can lead to improved DENOISEG performance. For the Fly Wing, DSB, and Mouse Nuclei data, we compare baseline results with DENOISEG results on the same data (n0) and with added synthetic noise (n10 and n20, see main text).

0.7 by computing the difference ($\Delta$) to $\alpha = 0.5$. Additionally we also compare to the Baseline and results that use (the a priori unknown) best $\alpha$. The best $\alpha$ for each trained network is found by a grid search for $\alpha \in \{0.1, 0.2, \ldots, 0.9\}$. Figure 6(a) shows that our proposed method is robust with respect to the choice of $\alpha$. Results for the other datasets showing similar trends are illustrated in Figure 7.

**Noisy Inputs Lead to Elevated Segmentation Performance.** Here we want to elaborate on the interesting observation we made in Figure 3: when additional noise is synthetically added to the raw data, the segmentation performance reaches higher AP and SEG scores, even though segmentation should be more difficult in the presence of noise. We investigate this phenomenon in Figure 6(b). We believe that in the absence of noise the denoising task can be solved trivially, preventing the regularizing effect that allows DENOISEG to cope with small amounts of training data.

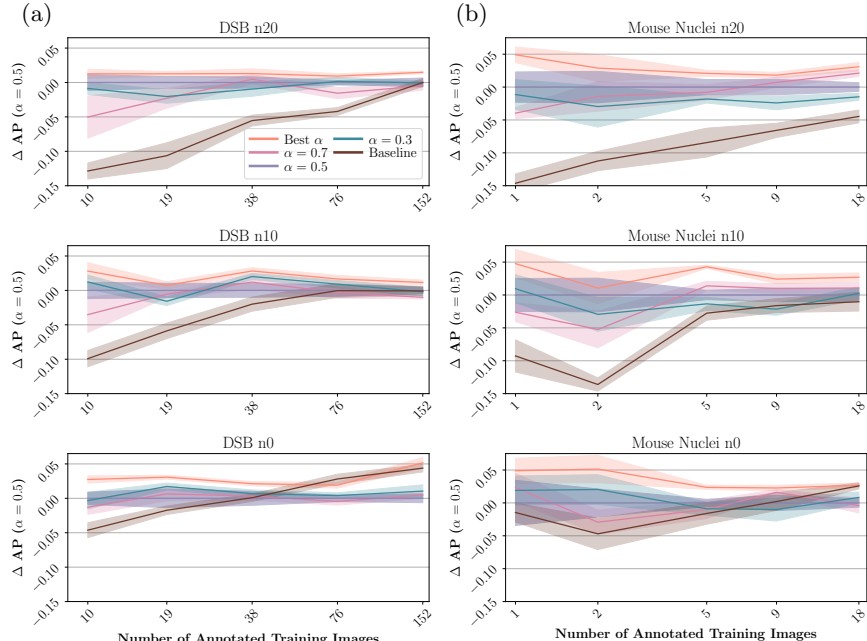

**Fig. 7.** We show the sensitivity of hyperparameter $\alpha$ for **(a)** Fly Wing and **(b)** Mouse Nuclei datasets by looking at the difference $\Delta$ of AP to $\alpha = 0.5$. Note that DENOISEG consistently improves results over the baseline for a broad range of hyperparameter $\alpha$ values.

**Evaluation of Denoising Performance.** Although we are not training DENOISEG networks for their denoising capabilities, it is interesting to know how their denoising predictions compare to dedicated denoising networks. Table 1 compares our denoising results with results obtained by NOISE2VOID [10]. It can be seen that co-learning segmentation is only marginally impeding the network's ability to denoise its inputs.

| Noise | DSB (GT for 10) | | Fly Wing (GT for 2) | | Mouse N. (GT for 1) | |
|---|---|---|---|---|---|---|
| | DENOISEG | NOISE2VOID | DENOISEG | NOISE2VOID | DENOISEG | NOISE2VOID |
| n10 | 37.57±0.07 | 38.01±0.05 | 33.12±0.01 | 33.16±0.01 | 37.42±0.10 | 37.86±0.01 |
| n20 | 35.38±0.08 | 35.53±0.02 | 30.45±0.20 | 30.72±0.01 | 34.21±0.19 | 34.59±0.01 |

**Table 1.** Comparing the denoising performance of DENOISEG and NOISE2VOID. Mean Peak Signal-to-Noise Ratio values (with ±1 SEM over 5 runs) are shown. Similar tables for DENOISEG results when more segmentation GT was available can be found online in the DENOISEG-Wiki.

## 4   Discussion

Here we have shown that (*i*) joint segmentation and self-supervised denoising leads to improved segmentation quality when only limited amounts of segmentation ground truth is available (Figures 2, 3, 4 and 5), (*ii*) the hyperparameter $\alpha$ is modulating the quality of segmentation results but leads to similarly good solutions for a broad range of values (Figures 6(a), 7), and (*iii*) results on input data that are subject to a certain amount of intrinsic or synthetically added noise lead to better segmentations than DenoiSeg trained on essentially noise-free raw data (Figure 6(b)).

We reason that the success of our proposed method originates from the fact that similar "skills" are required for denoising and segmentation. The segmentation task can profit from denoising, and compared to [16], performs even better when jointly trained within the same network. When a low number of annotated images are available, denoising is guiding the training and the features learned from this task, in turn, facilitate segmentation.

We believe that DenoiSeg offers a viable way to enable the learning of dense segmentations when only a very limited amount of segmentation labels are available, effectively making DenoiSeg applicable in cases where other methods are not. We also show that the amount of required training data can be so little, even ad-hoc label generation by human users is a valid possibility, expanding the practical applicability of our proposed method manyfold.

**Acknowledgments.** The authors would like to acknowledge ************ ***** and ************* from ******* for fly wing data, ************ and ****** *********** from ******* for mouse nuclei data and the Scientific Computing Facility at ******* for giving us access to their HPC cluster.

# Appendices

## A    DSB Results with Increasingly Many GT labels

The DSB dataset we have used offers the possibility to run DENOISEG with arbitrary many segmentation GT labels. While DENOISEG is intended in cases where the amount of such labels is very limited, in Figure 8 we plot the segmentation results of DENOISEG, the sequential baseline, as well as the baseline as defined in the main text.

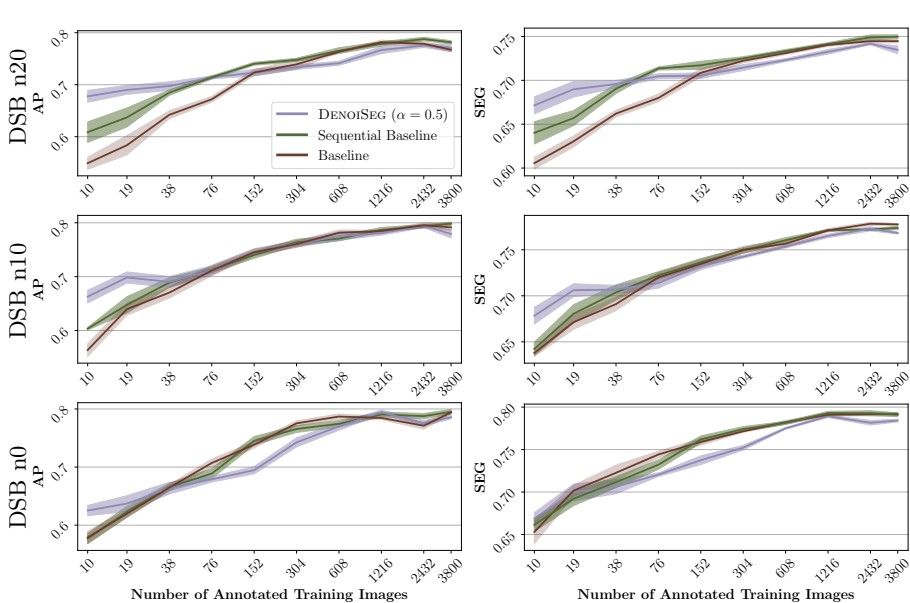

**Fig. 8.** Extended version of Figure 4. Results for DSB n0, n10 and n20, evaluated with Average Precision (AP) [21] and SEG-Score [24]. DENOISEG outperforms both baseline methods, mainly when only limited segmentation ground truth is available. Note that the advantage of our proposed method for this dataset is at least partially compromised when the image data is not noisy (row 3).

As expected, With additional labels, the advantage of also seeing noisy images decreases, leading to similarly good results for all compared methods. It is still reassuring to see that the performance of DENOISEG is still essentially on par with the results of a vanilla U-NET that does not perform the joint training we propose.

## B    Our Baseline *vs.* Vanilla 3-class U-Net

The baseline method we used in this work is, as explained in the main text, a DENOISEG network with $\alpha$ being set to 0. This is, in fact, very similar to using a vanilla 3-class U-NET. While we are still feeding noisy images, we are not backpropagating any denoising loss, meaning that only the data for which segmentation labels exist will contribute to the training. The one difference is, that some of the hyperparameters (number of epochs, adaptation of learning rate, *etc.*) will slightly diverge in these two baseline setups. Figure 9 shows that these subtle differences are in fact not making any practical differences.

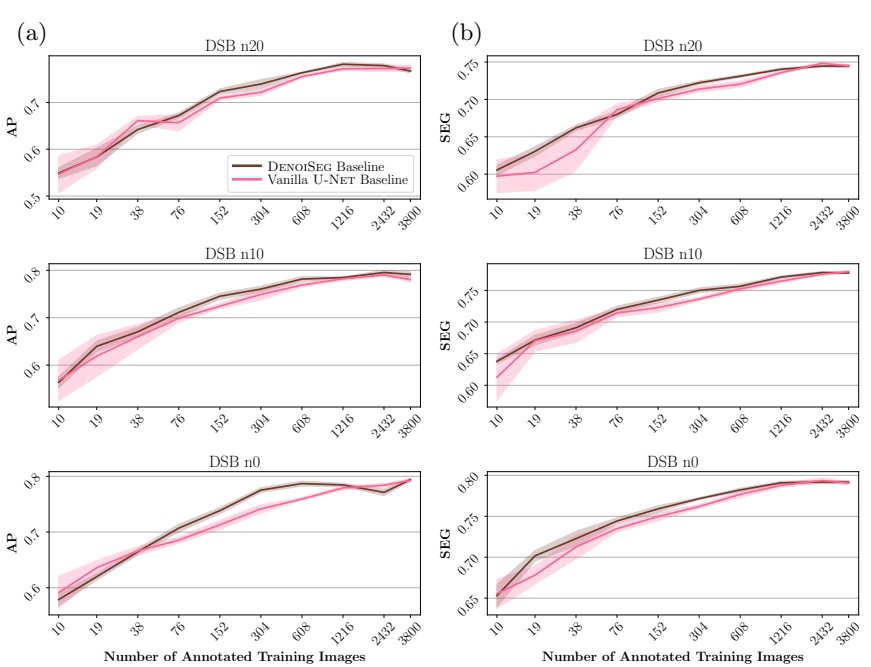

**Fig. 9.** Comparison of vanilla U-Net with our DENOISEG $\alpha = 0$ baseline for DSB datasets. Our DENOISEG $\alpha = 0$ baseline is at least as good or better than the vanilla U-net baseline both in terms of Average Precision (AP) [21] and SEG-Score [24] metrics. Hence, we establish a stronger baseline with DENOISEG $\alpha = 0$ and measure our performance against this baseline (see Figure 8, Figure 3 and Figure 5.)

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
