# OpenReview forum: "DenoiSeg: Joint Denoising and Segmentation"
_thecvf.com/ECCV/2020/Workshop/BIC — BIC 2020 Oral_

### Official Review · AnonReviewer2 · 2020-07-31
**DenoiSeg: Joint Denoising and Segmentation**

**Rating:** 9
**Confidence:** 4

**Review:**

Summary
This paper addresses the problem of image segmentation and denoising when a very few training segmentation masks are available. The problem is motivated by the need to train neural networks with a high capacity (millions of coefficients) while having only tens of ground truth segmentation masks. The authors approach the problem by combining the self-supervised denoising task with the segmentation task in one neural network optimized using a joint loss. The joint loss is a weighted contribution of denoising and segmentation loss contributions. The authors develop the joint denoising and segmentation framework as an extension to the Noise2Void work accessible at https://openaccess.thecvf.com/content_CVPR_2019/papers/Krull_Noise2Void_-_Learning_Denoising_From_Single_Noisy_Images_CVPR_2019_paper.pdf

Strengths:
The practical value of training a segmentation model with very few ground truth segmentation masks is very high.
The novelty lies in formulating a joint loss and delivering denoised images as well as segmentation masks.


Weaknesses:
The paper is missing an assumption paragraph which is misleading for a reader who would like to use this technique.  For example, one of the assumptions is the i.i.d. property of the noise. Another assumption is that the very few ground truth segmentation masks must be representative of the dataset. The authors showed the performance on three datasets that have spatially distributed very similar pattern/content and thus sampling is very easy.

Comments:
How did the authors decide on the size of the blind spot patches? The paper focused on Noise2Void is using patches of 64 x 64 pixels while this work is using patches of 128 x 128 pixels.
Lines 249 – 257: The authors refer to patches and then to images. Please, verify the terminology
Lines 314, 345-350: It is not clear how delta is computed. Is it AP(alpha=0.2) – AP(alpha=0.5) or AP(alpha=0.2) – AP(alpha=0.7) compared to AP(alpha=0.5)? Please, clarify. I could not follow the Figure 6 vertical axis (you might include an equation for delta).


**Reviews Visibility:**

I agree that my anonymized review is made publicly visible, if the submission is accepted.

---

### Decision · Program_Chairs · 2020-07-31

Accept (Oral)